# Integrated Transcriptome and sRNAome Analysis Reveals the Molecular Mechanisms of *Piriformospora indica*-Mediated Resistance to *Fusarium* Wilt in Banana

**DOI:** 10.3390/ijms252212446

**Published:** 2024-11-20

**Authors:** Junru Wang, Bin Wang, Junmei Huang, Shuai Yang, Huan Mei, Youfeng Jiang, Yacong Hou, Jun Peng, Chunzhen Cheng, Hua Li, Peitao Lü

**Affiliations:** 1College of Horticulture, Center for Plant Metabolomics, Haixia Institute of Science and Technology, Fujian Agriculture and Forestry University, Fuzhou 350002, China; 2National Key Laboratory for Tropical Crop Breeding, Institute of Tropical Bioscience and Biotechnology, Chinese Academy of Tropical Agricultural Sciences, Sanya 572024, China; 3College of Horticulture, Shanxi Agricultural University, Jinzhong 030801, China

**Keywords:** *Piriformospora indica*, banana, *Fusarium oxysporum*, transcriptome, miRNA

## Abstract

Bananas (*Musa* spp.) are among the most important fruit and staple food crops globally, holding a significant strategic position in food security in tropical and subtropical regions. However, the industry is grappling with a significant threat from *Fusarium* wilt, a disease incited by *Fusarium oxysporum* f. sp. *cubense* (Foc). In this study, we explored the potential of *Piriformospora indica* (Pi), a mycorrhizal fungus renowned for bolstering plant resilience and nutrient assimilation, to fortify bananas against this devastating disease. Through a meticulous comparative analysis of mRNA and miRNA expression in control, Foc-inoculated, Pi-colonized, and Pi-colonized followed by Foc-inoculated plants via transcriptome and sRNAome, we uncovered a significant enrichment of differentially expressed genes (DEGs) and DE miRNAs in pathways associated with plant growth and development, glutathione metabolism, and stress response. Our findings suggest that *P. indica* plays a pivotal role in bolstering banana resistance to Foc. We propose that *P. indica* modulates the expression of key genes, such as *glutathione S-transferase* (*GST*), and transcription factors (TFs), including TCP, through miRNAs, thus augmenting the plant’s defensive capabilities. This study offers novel perspectives on harnessing *P. indica* for the management of banana wilt disease.

## 1. Introduction

Bananas (*Musa* spp.) are not only globally renowned for their fruit but also serve as a vital food crop [1,2,3], exerting a profound impact on the economic well-being of tropical and subtropical regions [4]. Despite their importance, the banana industry is currently under siege by *Fusarium* wilt, a disease triggered by *Fusarium oxysporum* f. sp. *Cubense* (Foc) [5,6]. Commonly referred to as Panama disease or yellow leaf disease, this devastating condition obstructs the vascular system of bananas, impeding the essential flow of nutrients and water, ultimately culminating in plant mortality [7,8]. While various strategies such as physical [9], agricultural [10,11], biological [12,13] and resistance breeding [14] have been deployed to combat banana wilt disease, it is the advancement of sustainable biological control technologies and the development of resistant cultivars that hold the most promise for disease management. Central to biological control is the strategic utilization of biocontrol agents, which not only enhance the soil’s beneficial microbe population but also suppress the proliferation of wilt pathogens, aligning with the goals of environmental and ecological conservation. Consequently, biological control is emerging as a pivotal strategy in the effective management of banana wilt resistance [15,16].

*Piriformospora indica*, initially discovered in the roots of Indian desert shrubs [17], is a mycorrhizal fungus with an exceptional ability to colonize a broad spectrum of plant root systems [18]. Its agricultural applications have expanded significantly. Extensive research has demonstrated that *P. indica* not only stimulates the growth of diverse plants, including tobacco, corn, and rice, but also fortifies their defense against a plethora of biotic stresses such as fungal, bacterial, and viral attacks [19,20,21,22,23]. This enhancement is achieved through the induction of systemic disease resistance, bolstering antioxidant defenses, activation of signaling pathways, and modulation of plant hormone levels [24,25,26]. Furthermore, *P. indica* has been shown to accelerate plant development, reducing the vegetative growth phase and hastening the transition to the reproductive phase. This effect is likely attributed to its role in promoting robust root growth and optimizing nutrient uptake [27,28,29]. *P. indica* also plays a role in auxin synthesis, thereby fostering the growth of host plants like cabbage and *Arabidopsis* [30,31]. The ethylene-mediated promotion of plant growth, observed in *P. indica*-infected plants, remains an area ripe for further exploration. Evidence has accumulated that *P. indica* strengthens banana resistance to wilt disease [32], delaying symptom manifestation and enhancing the antioxidant enzyme activity and non-enzymatic antioxidant levels in banana roots. It also modulates the metabolism of glutathione-ascorbate and hormones, which are crucial for bolstering resistance to wilt [33]. Additionally, *P. indica* has been found to induce the expression of *GLP* genes in bananas and to reshape the rhizosphere microbial community, further fortifying resistance to this disease [34,35].

Glutathione S-transferase (GST) is a critical player in plant defense mechanisms, engaging in antioxidant responses alongside glutathione (GSH) to neutralize reactive oxygen species (ROS) produced under oxidative stress. This process is essential for plants’ resistance to biotic stresses. A wealth of research has demonstrated that *GST* genes are significantly induced by a variety of abiotic and biotic stressors [36,37]. In wild tomato plants harboring the Ol-1 resistance gene against powdery mildew caused by *Oidium neolycopersici*, a *GST* gene was upregulated, and *GST*-silenced plants showed susceptibility post-inoculation, indicating that *GST* is crucial for resistance to *O. neolycopersici* in tomatoes [38]. In Arabidopsis thaliana, GSH and indolyl glucosinolates have been established as pivotal components of the immune response [39]. *B. cinerea* infection in *Arabidopsis* resulted in the accumulation of catalase 3 and multiple *GSTs*, underscoring the antioxidant system’s role in fungal defense [40]. For Verticillium wilt, caused by the soil-borne necrotrophic fungus *V. dahliae*, the *GaGSTF9* gene in tree cotton (*Gossypium arboreum*) was identified as a pivotal regulator of resistance, modulating ROS and salicylic acid (SA) levels to affect plant disease resistance [41]. Analogously, infection by other pathogens, such as bacteria and viruses, induces oxidative stress that leads to the upregulation of *GST* genes. In heirloom tomatoes infected with *Pseudomonas syringae* pv., defense gene transcripts, including PR-1a, peroxidase, and *GST*, were upregulated in two resistant cultivars [42]. In peppers resistant to Capsicum chlorosis virus (CaCV), at least two *GST* genes were highly upregulated during lesion development [43]. WRKY transcription factor proteins are frequently associated with the regulation of antibacterial defense responses in plants [44,45]. Studies have suggested that WRKY TFs may play a role in the regulation of *GST* genes during fungal attacks on plants [46,47]. However, reports on the regulation of *GST* by other TFs during fungal infections are limited.

The TCP family represents a distinctive class of transcription factors within the plant kingdom, characterized by approximately 60 highly conserved amino acids within the TCP domain, situated at the protein’s N-terminus. This domain enables TCP proteins to bind to specific DNA sequences, thereby regulating gene expression. Taxonomically, the TCP family is bifurcated into two subfamilies, I and II, based on sequence variations within and beyond the TCP domain. Subfamily II is further branched into CIN (CINCINNATA) and CYC/TB1, with the latter being exclusive to flowering plants [48]. Initially recognized for their role in cell proliferation and expansion, TCP proteins have increasingly been implicated in a spectrum of plant developmental processes. These include, but are not limited to, seed germination, leaf maturation, floral formation, pollen development, and the intricate dance of the cell cycle [49]. In addition, TCP proteins play an important role in plant defense against pathogen invasion. They neutralize the inhibitory effect of the rps4-RLD1 protein, thereby enhancing the plant immune response to pathogens. Intriguingly, the absence of TCP proteins in the triple mutant *tcp8 tcp14 tcp15* results in enhanced susceptibility to *Pseudomonas syringae* pv. *tomato DC3000* (*Pst DC3000*), highlighting the significance of TCP proteins in the plant’s pathogen defense mechanism [50].

MicroRNAs (miRNAs) are a class of non-coding small RNA molecules in eukaryotes, typically 21–24 nt in length, which can regulate the expression of target genes by cutting target gene mRNA at the transcriptional level or by inhibiting their translation. miRNAs are not only involved in the regulation of plant growth and development [51,52,53,54] but also play a role in the response of plants to various abiotic stresses such as high temperature, high salinity, drought, low temperature, and heavy metals [55,56,57,58,59,60,61,62,63,64,65,66,67]. They regulate plant growth and development during stress by interacting with target genes to cope with various environmental stresses. In addition, an increasing number of studies have shown that miRNAs are widely involved in the regulation of plant immunity [58]. Plants can respond to the invasion of fungal and bacterial pathogens by regulating immune responses post-transcriptionally [59,60,61]. For example, *Arabidopsis* miR393 is the first miRNA found to trigger PTI (PAMP-triggered immunity) responses and can be induced by bacterial PAMP flagellin peptide flg22, negatively regulating the expression of auxin signaling pathway-related target genes, thereby inhibiting the plant auxin signaling pathway [62]. In addition to miR393, miR160 and miR167 are also induced by *Pst DC3000* and target auxin response factors to participate in plant immunity [63].

This study analyzed the impact of *P. indica* on the transcriptome and sRNAome of banana roots, discovering that genes in pathways such as glutathione metabolism play an important role in the *P. indica*-banana-*Foc* TR4 interaction. Co-expression network analysis showed that TCP proteins have co-expression relationships with multiple *GSTs* and miRNAs. This result preliminarily reveals the molecular mechanism by which *P. indica* enhances banana resistance to wilt disease, providing possibilities for the application of *P. indica* in the biological control of wilt disease.

## 2. Results

### 2.1. P. indica Exhibits Alleviating Effects on Banana Foc TR4 Infection

Through infecting banana roots with Foc, the uncolonized roots of *P. indica* exhibited more severe disease symptoms compared to the colonized roots (Figure 1). The colonization of *P. indica* effectively inhibited Foc infection and mitigated the degree of root browning (Figure 1A). To elucidate the mechanism of action of *P. indica* during the infection of banana by the wilt pathogen, RNA-seq analysis was conducted on four groups of banana samples under different treatment conditions: control (CK), *Foc* TR4 inoculation (Foc), *P. indica* colonization (Pi), and *P. indica* colonization followed by *Foc* TR4 inoculation (PF). Data analysis revealed that the average alignment rate of the four groups was 77.36%, with a Q30 of over 89% for each sample (Table 1). Pearson correlation matrix and PCA analysis were employed to filter out disparate samples, ensuring that the retained samples with high reproducibility would guarantee the credibility of subsequent analyses (Appendix A).

Differential expression analysis of the transcriptome data from the four groups of banana samples under different treatment conditions (Appendix A) identified 4381 and 926 significantly upregulated DEGs, as well as 3083 and 2108 significantly downregulated DEGs in the CK-vs.-Foc and Pi-vs.-PF comparisons, respectively (Figure 1B). To further verify the reliability of the RNA-seq data, 16 randomly selected genes with differential expression were subjected to qRT-PCR validation, and the results were consistent with the sequencing data, further confirming the reliability of the sequencing results (Figure 2). Among these, 4175 DEGs were specifically induced during *Foc* TR4 inoculation, while 720 DEGs were specifically induced by *Foc* TR4 after *P. indica* colonization. Additionally, 206 and 424 DEGs were commonly and significantly upregulated or downregulated in both Foc and PF groups relative to their respective control groups (Figure 1C). Notably, Gene Ontology (GO) enrichment analysis revealed that the 424 genes whose expression was simultaneously repressed in Foc and PF groups were significantly enriched in pathways related to nitrogen transport and phloem transport, such as response to nitrate, regulation of nitrogen utilization, nitrate transport, and phloem transport, as well as in pathways related to flavonoid/phenylpropanoid and brassinosteroid biosynthesis, which are associated with stress resistance. This suggests that *Foc* TR4 infection may affect the growth and development of banana plants by impacting nitrogen fixation and nutrient transport in the roots, and that flavonoid metabolism and brassinosteroid-related pathways are also suppressed during this process. Interestingly, the 4175 DEGs that were specifically induced in Foc group were significantly enriched in stress-responsive pathways, such as hyperosmotic salinity response, respiratory burst involved in defense response, response to jasmonic acid, jasmonic acid biosynthetic process, response to fungus, salicylic acid-mediated signaling pathway, defense response by callose deposition, response to mechanical stimulus, negative regulation of programmed cell death, systemic acquired resistance, and negative regulation of defense response. In contrast, the 720 DEGs that were specifically induced in the PF group were significantly enriched in response to hydrogen peroxide and glutathione metabolic process pathways (Figure 1D). This indicates that although *P. indica* still experiences the suppression of root nitrogen fixation and phloem nutrient transport during *Foc* TR4 infection, it can enhance plant resistance to pathogens by regulating glutathione metabolism.

### 2.2. P. indica Enhances Plant Resistance to Pathogens by Affecting GST Activity

To further clarify the alleviating mechanism of *P. indica* during the infection of banana by *Foc* TR4, hierarchical cluster analysis was performed on 8937 DEGs responsive to *Foc* TR4. The results revealed that the DEGs were distinctly divided into six modules. Specifically, modules C1 (1438), C2 (3248), C6 (1651), and C3 (1363) were specifically highly expressed in CK, Foc, Pi, and PF samples, respectively, while module C4 (578) was highly expressed in both Foc and Pi samples, and module C5 (659) was highly expressed in both Foc and PF samples (Figure 3A). GO analysis of the DEGs across six distinct modules revealed that those in module C2, which were significantly highly expressed during Foc treatment, were notably enriched in pathways associated with stress responses, including hyperosmotic salinity response, respiratory burst involved in defense response, jasmonic acid biosynthetic process, response to jasmonic acid, negative regulation of programmed cell death, response to mechanical stimulus, response to fungus, defense response by callose deposition, salicylic acid-mediated signaling pathway, systemic acquired resistance, and negative regulation of defense response (Figure 3B). Interestingly, the 659 DEGs (module C5) that responded to Foc under both CK and Pi conditions were also significantly enriched in stress-related pathways, such as the regulation of jasmonic acid-mediated signaling pathway, hyperosmotic salinity response, jasmonic acid biosynthetic process, cytokinin-activated signaling pathway, negative regulation of MAP kinase activity, induced systemic resistance, and response to insecticide. In contrast, the 1363 DEGs (module C3) specifically induced in the PF group were significantly enriched in the regulation of growth and glutathione metabolic process (Appendix A), suggesting that these two pathways are involved in the process by which *P. indica* enhances plant resistance to *Foc* TR4.

Further analysis of the DEGs enriched in the regulation of growth and glutathione metabolic process pathways revealed that *ADCP1* (*Ma11_g04710*, *Ma02_g12430*), *GST* (*Ma10_g09500*, *Ma09_g26590*, *Ma09_g26580*, *Ma09_g26540*, *Ma05_g10110*, *Ma04_g36090*, *Ma09_g26530*, *Ma05_g10100*), *YLR126C* (*Ma06_g31460*, *Ma04_g18160*), *ING2-like* (*Ma01_g20540*), *NSA2* (*Ma00_g03000*), and *CYP* (*Ma03_g08450*) genes were specifically induced by *Foc* TR4 inoculation after *P. indica* colonization (Figure 3C). Additionally, a multitude of TFs were enriched in module C3, including AP2/ERF, bHLH, MYB, HB, C2H2, LOB, NAC, B3, GRF, MADS, WRKY, and TCP families (Figure 3D,E, Appendix A). These results suggest that *P. indica* may enhance the antioxidant capacity of banana plants by regulating the expression of *GST* genes, thereby increasing their resistance to *Foc* TR4, and that these TFs may be involved in the regulation of these key genes.

### 2.3. WGCNA Analysis and Construction of Co-Expression Networks

To uncover genes linked to *P. indica*’s role in bolstering banana resistance to *Fusarium* wilt, we conducted a Weighted Gene Co-Expression Network Analysis (WGCNA) on the entire gene set, yielding seven distinct expression modules (Figure 4A,B and Appendix A). The MEsaddlebrown module, the largest one, encompassing 10,380 genes, exhibited the closest expression pattern to module C3, and the MEdarkgreen module with 6766 genes showed the highest congruence with C2′s expression trend (a Pearson correlation coefficient greater than 0.8, and significance less than 0.05) (Figure 3A and Figure 4B and Appendix A). Following in size were MEindianred, MEantiquewhite4, MEfirebrick4, and MEdarkolivegreen4, comprising 4085, 2382, 1523, and 930 genes, respectively. An additional 51 genes unclassified into any module were assigned to a gray module. To ascertain the co-expression similarities among modules, we determined the eigengenes for each and performed cluster analysis, revealing that six modules primarily clustered into two groups, with MEsaddlebrown and MEindianred displaying the greatest similarity (Appendix A). We identified a shared subset of 1199 genes present in both the MEsaddlebrown and C3 modules (Figure 4C), and 3009 genes present in both the MEdarkgreen and C2 modules (Appendix A). Through GO analysis of the MEsaddlebrown and C3 module, as well as the MEdarkgreen and C2 modules, we found that their enrichment results are similar to those of the individual C3 and C2 modules (Figure 3B and Figure 4D and Appendix A), indicating the functional consistency between the MEsaddlebrown and MEdarkgreen modules with the C3 and C2 modules, which provides assurance for the subsequent construction of the co-expression network. Building on this, and using the C3 module genes as a reference, which related to the process of increasing plant resistance to *Foc* TR4 by *P. indica*, we constructed a co-expression network for these 1199 genes. This network encompassed genes implicated in growth regulation and glutathione metabolism, including *GST* (*Ma10_g09500*, *Ma09_g26530*, *Ma05_g10100*), *CYP* (*Ma03_g08450*), *ACBD3* (*Ma04_g26630*), *INO80* (*Ma06_g23090*), and the *Arabidopsis AT1G50140* gene homolog, *Ma11_g14410*, which in *Arabidopsis* is known to encode a P-loop nucleoside triphosphate hydrolase superfamily protein (which possess ATP hydrolysis activity) involved in energy transport [64]. These genes served as network hubs, with only TFs that showed co-expression relationships being included. The TFs were stratified according to the number of connections, or degree, with the core layer having the highest degree (Figure 4E).

### 2.4. miRNA-Mediated Regulation of GST in P. indica-Enhanced Banana Resistance to Foc TR4

To elucidate the contribution of miRNAs to the *P. indica*-induced enhancement of banana resistance against *Fusarium* wilt, we conducted sequencing on sRNA libraries from 12 samples (Table 2). Following the exclusion of inconsistent samples through Pearson correlation and PCA analysis, we secured a dataset of high reproducibility, foundational for reliable subsequent analyses (Appendix A). The rRNA predominates across all treatments in all sRNA categories (Figure 5A), with uridine being the most frequent at the initial base position (Figure 5B). Alignment to the miRbase database identified 185 known miRNAs across 37 families, with the miR396 family being the most numerous (21 members), followed by miR156 (17 members) and miR171 (16 members). The miR169, miR166, miR172, and miR319 families each contained over 10 members, while the miR160, miR393, miR167, miR164, miR529, miR399, miR845, and miR159 families ranged from 4 to 9 members. The miR157, miR162, miR168, miR398, miR408, and miR390 families each had three members, the miR397, miR535, and miR395 families had two members each, and the remaining families consisted of a single member (Figure 5C). The CK, Foc, Pi, and PF groups yielded 158, 171, 169, and 169 known miRNAs, respectively.

By analyzing target genes of all miRNAs that we have identified, we found that they enriched growth and development, as well as stress response pathways, including the positive regulation of development, heterochronic process, leaf morphogenesis, response to bacterium, regulation of anthocyanin biosynthetic process, regulation of secondary metabolite biosynthetic process, etc. This indicates that miRNAs are closely related to plant growth and development, as well as stress responses (Appendix A). Analysis of Differentially Expressed miRNAs (DE miRNAs) in the CK-vs.-Foc and Pi-vs.-PF comparison groups revealed 69 and 24 known miRNAs, along with 51 and 54 novel miRNAs, respectively. Within the CK-vs.-Foc group, we observed an upregulation of 40 known miRNAs and a downregulation of 29, whereas the Pi-vs.-PF group exhibited the upregulation of 10 and the downregulation of 14 miRNAs (Figure 6A, Appendix A). Collectively, 94 known DE miRNAs were identified across both groups, with 69 (40 upregulated, 29 downregulated) of them in the CK-vs.-Foc and 25 (10 upregulated, 15 downregulated) in the Pi-vs.-PF (Figure 6B). Intriguingly, four and seven DE miRNAs were commonly modulated in response to both *Foc* TR4 treatment alone and following *P. indica* colonization. Specifically, 36 and 6 miRNAs were induced, while 22 and 17 were suppressed in each group, respectively. Heatmap analysis revealed that, aside from miR167d_1 peaking with Pi treatment, the commonly downregulated DE miRNAs were most abundant in the untreated state. In the Pi-vs.-PF group, aside from miR160a-5p peaking under Pi treatment, the specifically suppressed miRNAs post-*P. indica* colonization and *Foc* TR4 infection showed significantly lower expression than with *Foc* TR4 infection alone (Figure 6C). This suggests a potential role for these miRNAs involved in the process of *P. indica* fortifying banana root resistance to *Foc* TR4 infection.

Following the prediction of miRNA target genes (Appendix A), we deliberately selected six established miRNAs for further examination using poly(A)miRNA-based qRT-PCR, which allowed us to assess their expression patterns in relation to their target genes (Figure 6D). The qRT-PCR data corroborated well with the sRNA sequencing findings. Post-*Foc* TR4 infection, a significant upregulation was observed for miR156h, miR160b, miR529b, miR160f-5, and miR393h, with respective fold changes of 2.5, 2.82, 3.88, 1.22, 3.85, and 1.43 compared to the control (CK). This upregulation underscores their potential roles in the defense response to *Foc* TR4. Correlative analysis between miRNAs and their targets disclosed pronounced negative correlations, thereby validating the sequencing data. Specifically, miR156a-5p showed an inverse relationship with *NBS-LRR*, miR156h with *TIFY*, miR160b with *UDP*, miR160f-5p with *ARF*, miR529b with *GT2*, and miR393h with *TIR1*. These correlations substantiate the regulatory interplay between these miRNAs and their targets in the context of *Foc* TR4 infection. Subsequent analysis identified 17 target genes that were specifically downregulated in the Pi-vs.-PF comparison group, encompassing key genes such as *GSTs* (*Ma04_g36090*, *Ma09_g26530*, *Ma09_g26580*, *Ma09_g26590*, *Ma05_g10100*, *Ma10_g09500*), *ACBD3* (*Ma04_g26630*), *YLR126C* (*Ma04_g18160*), *ADCP1* (*Ma11_g04710*), *NSA2* (*Ma00_g03000*), *INO80* (*Ma06_g23090*), *YLR126C* (*Ma06_g31460*), and *Ma11_g14410*. Additionally, *TCP* TFs (*Ma04_g38090*, *Ma03_g32580*, *Ma10_g14460*) and *SBP* (*Ma03_g10910*) were among the identified targets. This comprehensive analysis highlights the intricate regulatory network where miRNAs may directly or indirectly modulate the expression of genes, such as *GST*s, influencing the banana’s resistance to *Foc* TR4.

GO analysis of the target genes of 17 Pi-vs.-PF specifically downregulated DE miRNAs (Figure 6E) revealed significant enrichment of miRNA target genes in pathways integral to plant growth and development. These included processes such as positive regulation of development, heterochronic timing, anther development, leaf morphogenesis, leaf shaping, regulation of the vegetative to reproductive phase transition timing, petal development, developmental growth, and root cap development. This enrichment pattern suggests that *P. indica* might modulate the expression of miRNAs, thereby activating target genes that promote plant growth and development. This activation could counteract the growth suppression induced by *Foc* TR4 infection in banana plants, particularly impacting leaf and anther development. Furthermore, these genes were notably enriched in pathways associated with the biosynthesis of secondary metabolites, including the regulation of anthocyanin biosynthetic processes and overall secondary metabolite biosynthesis. This finding implies that *P. indica* not only influences primary growth and development but also modulates the production of metabolic compounds that are crucial for plant defense and adaptation to environmental challenges.

The integrated analysis of DE miRNA target genes and WGCNA findings uncovered significant co-expression networks between the downregulated miRNA targets in the Pi-vs.-PF comparison group and genes pivotal to glutathione metabolism and developmental biosynthesis (Figure 6F). Notably, genes such as *GST* (*Ma09_g26530*, *Ma05_g10100*, *Ma10_g09500*), *ACBD3* (*Ma04_g26630*), *INO80* (*Ma06_g23090*), and *Ma11_g14410*, along with four transcription factor genes, were central to this network, as previously depicted in Figure 3C. The network analysis depicted in Figure 6F highlighted the TF *TCP* (*Ma03_g32580*) as a key hub with co-expression links to all key genes. It was followed by *TCP* (*Ma04_g38090*), *SBP* (*Ma03_g10910*), and *TCP* (*Ma10_g14460*), with their interactions mirroring those observed in Figure 3C. Specific miRNAs, including miR160g_1, miR396c-3p, and miR396e, were directly correlated with key genes *INO80* (*Ma06_g23090*), *Ma11_g14410*, and *ACBD3* (*Ma04_g26630*), respectively. Furthermore, miR319b_1, miR319a-3p, and miR529b exhibited co-expression relationships with transcription factors *TCP* (*Ma04_g38090*), *TCP* (*Ma03_g32580*), *TCP* (*Ma10_g14460*), and *SBP* (*Ma03_g10910*), respectively. These findings suggest a sophisticated regulatory role for miRNAs, potentially modulating the expression of genes such as glutathione antioxidant enzymes and TFs, which could bolster the plant’s defense mechanisms against pathogens. This intricate interplay between miRNAs and their targets provides a nuanced view of how post-transcriptional regulation can influence plant resistance to diseases like *Fusarium* wilt.

## 3. Discussion

Plants deploy a suite of hormones, including jasmonic acid (JA), ethylene (ET), salicylic acid (SA), and indole-3-acetic acid (IAA), in response to pathogen assault, thereby fortifying their disease resistance [65,66]. Particularly, the JA signaling pathway is instrumental in mounting resistance against necrotrophic pathogens like *Fusarium oxysporum* and *Botrytis cinerea* [67,68]. In our study, differentially expressed genes (DEGs) that were specifically upregulated in the CK-vs.-Foc comparison were notably enriched in pathways involved in the jasmonic acid biosynthetic process, response to jasmonic acid, and salicylic acid-mediated signaling pathway (Figure 1D). This suggests that when banana seedlings are challenged by *Foc* TR4, they trigger the synthesis and signaling pathways of JA and SA to combat fungal infection and initiate defense mechanisms against external threats. Consistent with previous research, the activation of plant PTI (PAMP-triggered immunity) is known to initiate a cascade involving calcium-dependent protein kinases (CDPKs) and mitogen-activated protein kinase (MAPK), which subsequently modulate downstream immune responses [69]. In the resistant banana cultivar ‘Guangjiao 9’, the expression of *MEKK3* is significantly upregulated following *Foc* TR4 infection, suggesting that the PTI response is activated, enhancing the plant’s defense against pathogen invasion through the induction of downstream signal transduction pathways [70]. The response of DEGs in module Cluster 5 in both Foc and PF groups is noteworthy. These DEGs were significantly enriched in pathways involved in the regulation of jasmonic acid-mediated signaling, the biosynthesis of jasmonic acid, and the negative regulation of MAP kinase activity, aligning with the findings from previous studies. This enrichment pattern underscores the complex regulatory mechanisms at play in the banana’s defense response to *Fusarium* wilt. In the C2 module, the majority of the genes induced by *Foc* TR4 were not elicited following the treatment with *P. indica*. The DEGs within this module were enriched in pathways associated with stress responses, such as the respiratory burst involved in the defense response, the response to jasmonic acid, the jasmonic acid biosynthetic process, and the response to fungus. The expression patterns of genes in the C2 and C3 modules were nearly contrary. In the C3 module, the majority of the genes downregulated by Foc were induced after the treatment with *P. indica*. We focused on the DEGs specifically induced by the reinfection with *Foc* TR4 after the colonization by *P. indica*, which were significantly enriched in the regulation of growth and glutathione metabolic process pathways.

*P. indica*’s capacity to bolster plant resistance against a spectrum of pathogens is well documented. It has been demonstrated that *P. indica* can augment the antioxidant defense mechanisms in plants and stimulate the expression of genes associated with disease resistance, thereby providing protection against pathogen invasion. Evidence suggests that *P. indica* fortifies the antioxidant defense system, as seen in its ability to render chickpea plants resistant to gray mold [71]. Similarly, it has been observed that wheat plants, when colonized by *P. indica*, exhibit heightened basal immunity against crown rot. This enhancement is attributed to the synergistic effects of polyamines and nitric oxide, which modulate hydrogen peroxide levels and activate antioxidant enzymes such as POD and CAT. These actions lead to callose deposition, a critical component of plant resistance induced by *P. indica* [72]. Further, *P. indica* has been shown to shield barley roots from the antioxidant capacity decline typically induced by *Fusarium culmorum*. Pre-inoculation with *P. indica* has been noted to effectively mitigate the pathogen-induced reduction in the ascorbate/glutathione ratio and glutathione content [73]. Moreover, *P. indica* contributes to increased rice resistance against sheath blight by diminishing H2O2 levels and elevating antioxidant enzyme activity [74]. Collectively, these findings indicate that *P. indica* colonization can invigorate the plant’s antioxidant enzymes, neutralize reactive oxygen species (ROS) within plant cells, and, by doing so, initiate defense responses that bolster plant resistance to pathogens.

In this study, our analysis of the transcriptome revealed that DEGs associated with *P. indica* colonization in *Foc* TR4-infected banana plants were prominently enriched in the glutathione metabolic process pathway. Consistent with this, both transcriptome and sRNA data indicated that DEGs and DE miRNA target genes, following PF treatment, were significantly enriched in pathways that regulate growth and development. We hypothesize that *P. indica* potentially modulates the growth and developmental trajectory of banana seedlings to counteract the detrimental effects of *Foc* TR4, concurrently boosting their antioxidant capacity through the regulation of glutathione metabolism. By delving into these pathways, we discovered that the structural genes in these pathway gene sets included *GST*, *CYP*, *ADCP1*, *ING2-like*, and other genes. GST is an essential enzyme in the plant antioxidant process, participating in plant stress responses, detoxification, and in the promotion of plant growth and development. It is involved in auxin transport, secondary metabolite metabolism, and catalyzes the metabolism of various toxic compounds [75,76]. The gene *CYP* (*Ma03_g08450*) encodes a protein belonging to the Non-A-Type CYP72 family, which is implicated in plant growth and development and stress responses [77]. Additionally, *ADCP1* (*Ma11_g04710*, *Ma02_g12430*) and *ING2-like* (*Ma01_g20540*) genes are related to chromatin regulation. These two genes in Arabidopsis are involved in different histone modifications, and *ING2-like* genes (*Ma01_g20540*) can also regulate DNA methylation levels [78,79]. Similarly, the target genes of miRNAs specifically downregulated in Pi-vs.-PF also encompass these genes, suggesting that they may comprehensively regulate the growth and development and stress responses of banana plants by influencing the epigenetic modification status and the expression levels of genes, such as *GST*, during the process in which *P. indica* enhances banana resistance to *Fusarium* wilt. Furthermore, among these target genes, *YLR126C* (*Ma06_g31460*, *Ma04_g18160*) encodes a glutaminase transferase [80], *NSA2* (*Ma00_g03000*) is a homolog of the ribosomal biogenesis protein NSA2, and *ACBD3* (*Ma04_g26630*) encodes acyl-CoA-binding domain-containing protein 3, which is involved in steroidogenesis, membrane transport, and viral/bacterial proliferation in infected host cells in mammals. Nevertheless, their functions in plants are less studied and remain ambiguous [81]. INO80 belongs to a subfamily of ATP-dependent chromatin remodeling complexes (CRCs) and plays a vital role in processes such as DNA repair, DNA replication, telomere maintenance, and chromosome segregation. INO80 seems to be involved in plant flowering, immune responses, and the regulation of miRNA expression, but the functional role and molecular mechanism of INO80 in plant defense are still poorly understood [82]. Subsequent co-expression network analysis revealed intricate co-expression relationships between these genes and TFs like bHLH and MYB. Notably, TCP proteins emerged as a potential target of miRNA regulation, suggesting that they play a role in modulating the expression of these key genes, thereby finetuning the plant’s defense and developmental responses.

As a beneficial symbiotic bacterium, *P. indica* is known to enhance banana resistance to wilt disease. Prior research has established that *P. indica* colonization can boost antioxidant enzyme activity and non-enzymatic antioxidant levels in banana roots, modulate ascorbate-glutathione metabolism, and influence hormone and fatty acid metabolism, thereby improving banana’s resilience to wilt disease. These findings have begun to shed light on the physiological and biochemical mechanisms underlying *P. indica*’s role in enhancing resistance. However, the molecular mechanisms remain largely elusive, and no comprehensive transcriptome studies have explored the interaction between *P. indica*, bananas, and *Foc* TR4. In this study, leveraging transcriptome and sRNA sequencing, we identified DEGs and DE miRNAs along with their target genes in response to *P. indica*’s induction of *Foc* TR4. Our analysis delved into the molecular mechanisms by which *P. indica* induces resistance to wilt disease in bananas. We propose a regulatory network involving miRNAs, transcription factors such as TCP, and structural genes like *GST*, *YLR126C*, *INO80*, and *NSA2*, which are implicated in plant stress response and developmental processes. This study offers novel insights into the molecular underpinnings of *P. indica*’s role in bolstering banana resistance to wilt disease.

## 4. Materials and Methods

### 4.1. Experimental Materials and Treatment

‘Tianbao’ banana cultivar seedlings were selected for uniformity and health, and then they were transplanted and allocated into two groups after a one-week acclimation period. The first group received irrigation with *P. indica* fermentation liquid, designated as the Pi group, while the second group was irrigated with an equivalent volume of PDB, serving as the control group (CK). After a two-week period, the establishment of *P. indica* colonization was assessed, and any plants that failed to demonstrate colonization were excluded from the Pi group. Thereafter, both the CK and the successfully colonized Pi groups were split into two subsets; one subset remained untreated, maintaining their CK and Pi statuses, while the other subset was inoculated with *Foc* TR4, resulting in the Foc (CK with *Foc* TR4 inoculation) and PF (Pi with *Foc* TR4 inoculation) groups.

Adhering to the sampling criteria set forth by Cheng et al. [83], we documented the onset of disease in the Foc group and pinpointed the initiation of wilt disease symptoms as the critical sampling juncture for sequencing. All sampling instruments were subjected to RNase-free and high-temperature sterilization processes. The roots were meticulously washed to remove soil, followed by grinding in liquid nitrogen. The ground root material was then stored in 10 mL centrifuge tubes at −80 °C for further analysis.

### 4.2. Cultivation and Inoculation of P. indica and Foc TR4 Strains

PDA medium was employed to cultivate *P. indica* and *Foc* TR4 in darkness at a temperature of 28 °C. Upon reaching a growth where the mycelium covered three-quarters of the PDA medium, 3–4 agar plugs were excised from the periphery of the *P. indica* colony and transferred to a triangular flask containing 300 mL of PDB medium. This flask was then incubated at 28 °C with a shaking speed of 200 rpm for three days to produce the *P. indica* fermentation liquid. The liquid was adjusted to a spore concentration of 1 × 10^6^/mL for root immersion, following the inoculation protocol described by Cheng et al. [33]. The banana root system was gently scarified with a sterile scalpel and immersed in the *P. indica* fermentation liquid for a period of 6 h. Subsequently, the seedlings were transplanted into pots and irrigated with the *P. indica* fermentation liquid in proximity to the root system, repeated every three days for a total of three applications.

For the *Foc* TR4 inoculation, after the mycelium had fully colonized the PDA medium, the agar was aseptically washed away, and the mycelium was passed through four layers of sterile gauze to filter and collect the spores. These spores were then centrifuged and resuspended in Hoagland nutrient solution to achieve a concentration of 1 × 10^7^/mL, ready for inoculation. The inoculation process adhered to the method detailed by Wang et al. [84], where the banana roots were cleansed and immersed in a 200 mL spore suspension of *Foc* TR4 at a concentration of 1 × 10^7^/mL for 72 h. Following this treatment, the plants were replanted into their original substrate, and the soaking liquid was applied adjacent to the root system (50 mL per plant). The plants were then positioned in a greenhouse for cultivation.

### 4.3. Real-Time Quantitative PCR (qRT-PCR) Analysis

Total RNA extraction from banana root samples was conducted utilizing the Tiangen kit protocol (Tiangen, Beijing, China). Prior to library construction, the total RNA was meticulously assessed for concentration, purity, and integrity.

For qRT-PCR, cDNA was synthesized using the PrimeScript™ RT kit from TransGen (Beijing, China). The qRT-PCR procedure adhered to the methodology outlined by Cheng et al. [35] and was executed on the LightCycler 480 system (Roche, Basel, Switzerland). Clathrin adaptor complexes (CACs) served as the reference gene for examining the expression of 16 target genes. All quantification primers were meticulously designed utilizing the Primer 3 software (version 0.4.0), and detailed primer sequences are presented in Appendix A.

The TransScript miRNA First-Strand cDNA Synthesis SuperMix (TransGen, Beijing, China) facilitated miRNA reverse transcription. The expression of 8 miRNAs and their corresponding target genes was validated via real-time quantitative PCR. This included 4 miRNAs that exhibited differential expression in the CK-vs.-Foc group (miR156a-5p with *NBS-LRR*, miR156h with *TIFY*, miR160b with *UDP*, and miR160f-5p with *ARF*), as well as 2 miRNAs that displayed differential expression in both the Pi-vs.-PF and Foc-vs.-PF groups (miR529b with *GT2* and miR393h with *TIR1*). The primers utilized are detailed in Appendix A. The 3′ end of sRNA was polyadenylated using the Poly(A) tailing kit (Thermo Scientific, Pittsburgh, PA, USA), with U6 as the internal reference gene. The relative expression levels of each gene were determined by employing the 2^−ΔΔCT^ quantification method.

### 4.4. Transcriptome Analysis and Identification of DEGs

The raw data were refined for quality using trim_galore software (version 0.6.7) [85], ensuring the removal of adapters, reads with greater than 10% unknown base N content, and low-quality sequences. The resultant clean reads were aligned to the banana genome utilizing hisat2 software (version 2.1.0) [86]. Subsequently, the R package Rsubread (version 2.8.1) [87] with the featureCounts function was employed to quantify the reads mapped to each gene, represented as counts. These counts were normalized employing the TPM (transcripts per million mapped reads) method.

Differential expression analysis was conducted on the counts matrix using DESeq2 software (version 1.34.0) [88], which incorporates an internal normalization method with rlog transformation. The analysis yielded a *q*-value, log_2_ fold change (log_2_FC), and the mean of TPM (mean [TPM]) for the screening of DEGs. The criteria for DEGs were set as follows: *q*-value ≤ 0.01, |log_2_FC| and mean (TPM) ≥ 1. For GO analysis, the R package clusterProfiler (version 4.2.2) [89] was utilized, while VennDiagram (version 0.6.5) and pheatmap (version 1.0.12) were employed for the creation of Venn diagrams and heatmaps, respectively. Additionally, ggplot2 (version 3.3.6) [90] was applied for the generation of other graphical representations.

Weighted Gene Co-Expression Network Analysis (WGCNA) was executed with the R package WGCNA (version 1.72.5) [91], using the mean (TPM) ≥ 1 of 26,117 genes’ TPM expression matrix as the foundational data input. The analysis parameters were set with a power value of 28 and a minModuleSize of 100. Modules identified by WGCNA were cross-referenced with the hierarchical clustering results of DEGs that exhibited similar expression patterns. Genes exhibiting differential expression within the WGCNA modules were selected for network construction. Network visualization was accomplished using Cytoscape software (version 3.9.1) [92].

### 4.5. sRNA Annotation and Identification of DE miRNAs

The preprocessing of raw sRNA data involved the removal of adapter contaminants and poor-quality sequences, achieved using cutadapt (version 2.8) [93] and fastp (version 0.20.0) software. Thereafter, the refined clean data were aligned to the banana genome with the bowtie software (version 1.2.3) [94]. The annotation of sRNAs was systematically categorized into various RNA categories. All small RNAs were aligned against the miRNA precursor database, ncRNA database, repeat database, and the exon and intron regions of the genome of the species that were constructed. The classification and annotation results were prioritized in the order of known miRNAs > ncRNAs > repeats > exons > introns, and sRNAs were annotated by traversing this priority sequence. Information on miRNA families was sourced from the Rfam database [95]. The prediction of novel miRNAs and the identification of their target genes were conducted using miRDP2 software (version 1.1.4) [96] and TargetFinder, along with psRobot (version 1.2) [97]. The expression levels of miRNAs were standardized using the TPM (transcripts per million) method. DESeq2 software (version 1.34.0) was then utilized to calculate the differential expression of miRNAs. miRNAs exhibiting a fold change of ≥2 and a *q*-value of ≤0.01 were designated as DE miRNAs, indicating significant changes in their expression profiles.

## Figures and Tables

**Figure 1 ijms-25-12446-f001:**
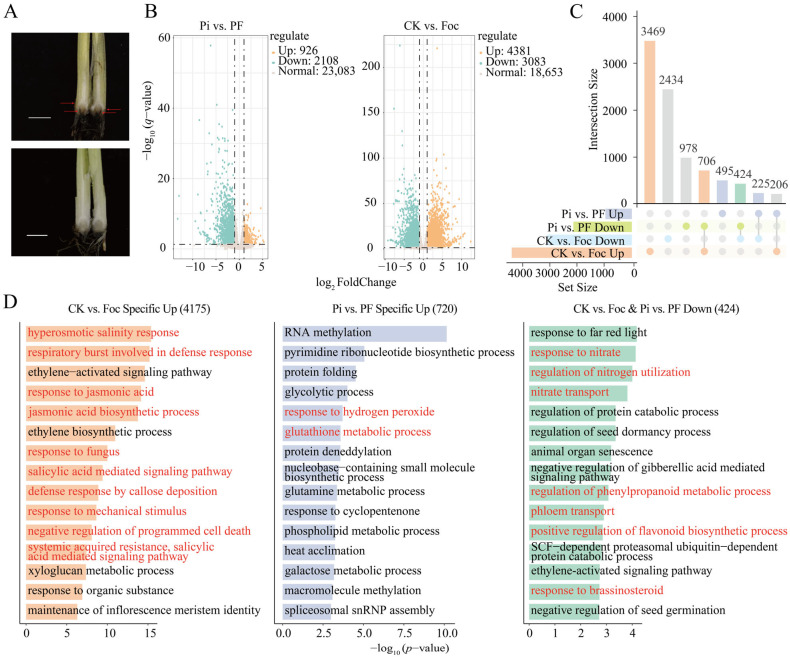
DEG statistics and GO enrichment analysis in Pi-vs.-PF and CK-vs.-Foc comparisons. (**A**) Phenotypic manifestations of *Foc* TR4 infection in the corms of banana seedling. The top panel displays non-inoculated bananas, while the bottom panel shows those inoculated with *P. indica*. Red arrows represent the diseased zones. Bar = 1 cm. (**B**) Volcano plots depicting downregulated DEGs as green dots and upregulated DEGs as orange dots in two comparative analyses. (**C**) UpSet plot displaying the concordant and unique regulation of DEGs between the two comparisons. (**D**) GO enrichment analysis highlighting the biological pathways significantly associated with specifically upregulated and co-downregulated DEGs identified in Pi-vs.-PF and CK-vs.-Foc comparisons. Red content shows pathways related to stress response or growth and development mentioned in the article.

**Figure 2 ijms-25-12446-f002:**
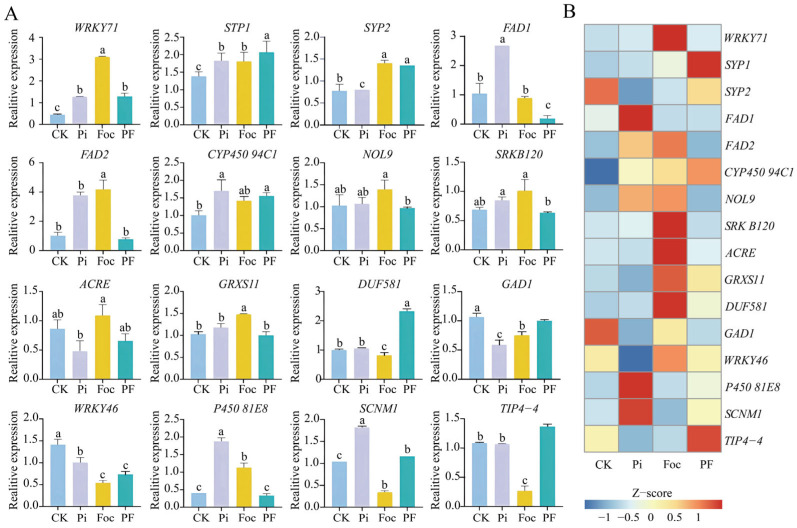
qRT-PCR verification results of differentially expressed genes. The relative expression histogram is displayed on the left, accompanied by significance analysis conducted using the ANOVA method (*p* < 0.05) (**A**), and the heatmap is on the right (**B**).

**Figure 3 ijms-25-12446-f003:**
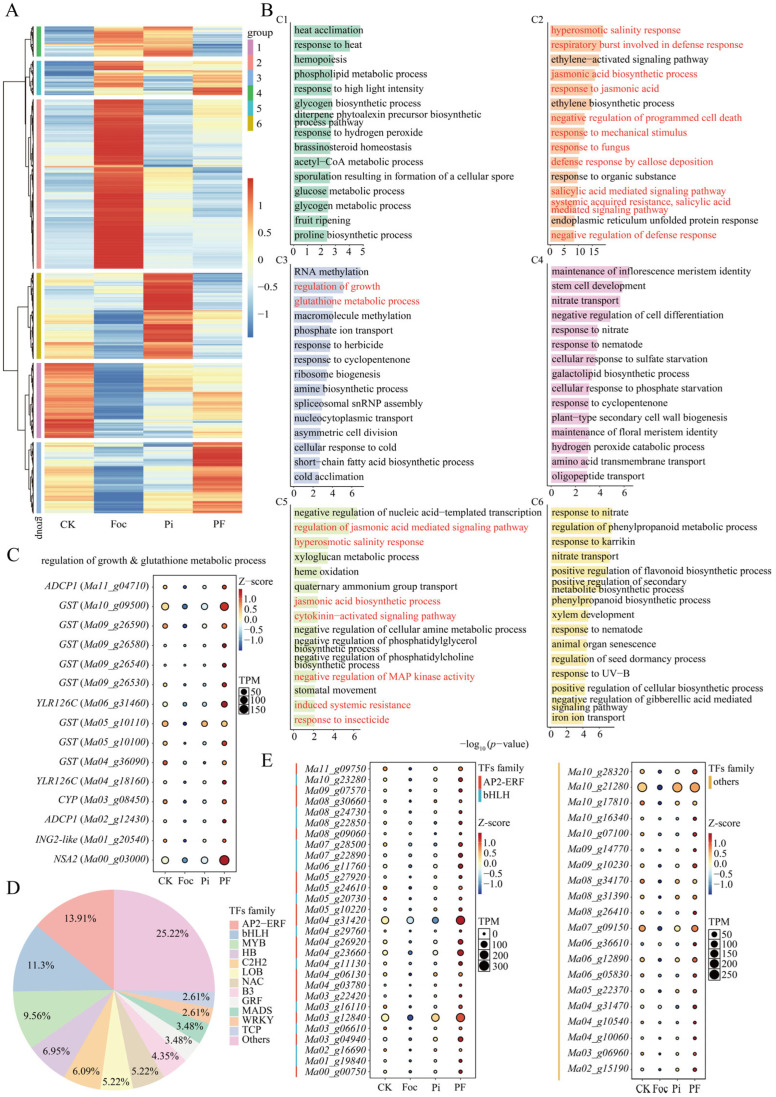
Hierarchical cluster and GO enrichment analysis of 13,801 Foc-responsive DEGs. (**A**) Hierarchical cluster analysis of 13,801 DEGs, resulting in the identification of 6 distinct modules. (**B**) GO analysis for the 6 modules, with a focus on the biological process (BP) enrichment results. The red font denotes the pathways related to stress response or growth and development mentioned in C2, C3 or C5 modules. (**C**) Bubble chart representing the expression levels of structural genes in Cluster 3, involved in growth regulation and glutathione metabolism. (**D**) Pie chart illustrates the proportion of transcription factors within the C3 module. (**E**) Bubble chart showing the expression of *AP2-ERF*, *bHLH*, and other transcription factor families in Cluster 3.

**Figure 4 ijms-25-12446-f004:**
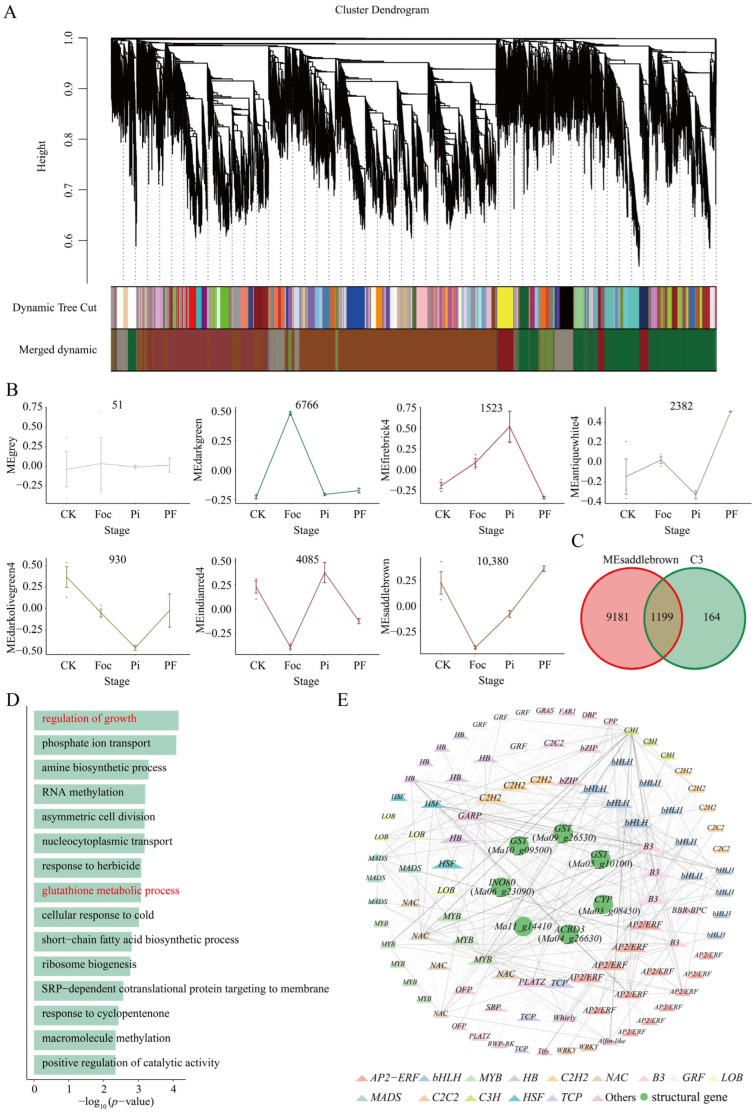
WGCNA co-expression analysis and gene network construction. (**A**) Dendrogram of the WGCNA cluster analysis, genes are categorized into diverse modules and displayed in various colors, which serve as the names for these 7 corresponding modules (**B**). (**C**) Venn diagram representing the gene intersection between hierarchical Cluster 3 and the MEsaddlebrown module, totaling 1199 genes. (**D**) GO analysis of 1199 genes which only focus on the biological process (BP) enrichment results. The red content is the same as pathways marked in the C3 module in Figure 3B. (**E**) Co-expression network of the 1199 genes, highlighting structural genes at the core and surrounding transcription factors.

**Figure 5 ijms-25-12446-f005:**
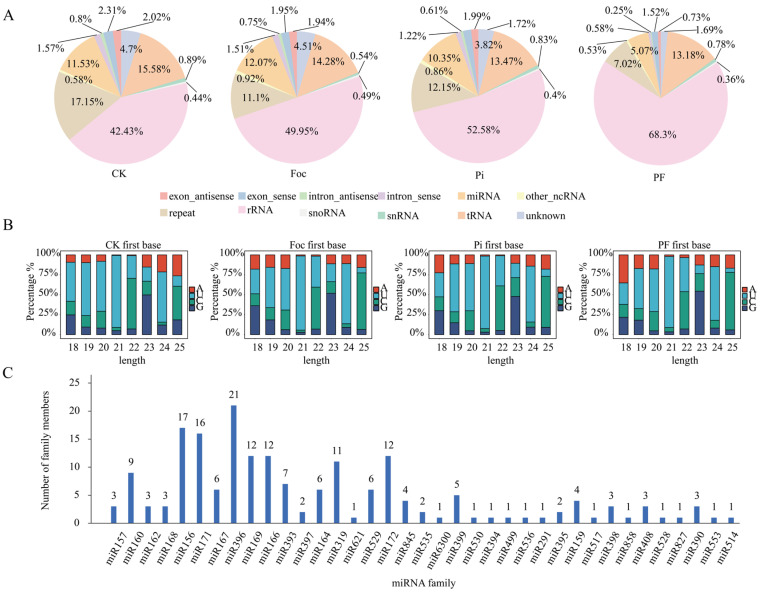
sRNA analysis of 4 treatments. (**A**) The proportion of different types of sRNA in 4 treatments. (**B**) The first base distribution of miRNA. (**C**) Number of miRNA family members in banana root.

**Figure 6 ijms-25-12446-f006:**
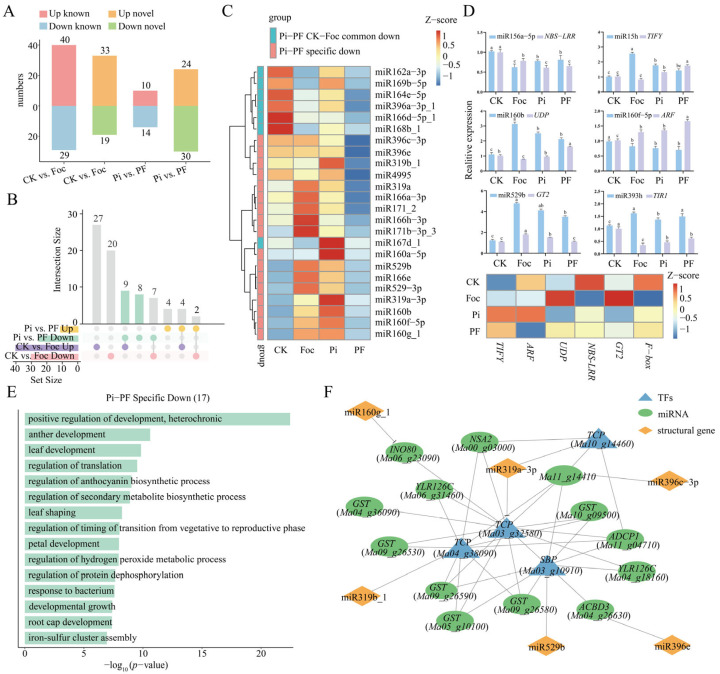
Analysis of DE miRNAs in Pi-vs.-PF and CK-vs.-Foc comparisons and network construction. (**A**) Quantification of downregulated known, downregulated novel, upregulated known, and upregulated novel DE miRNAs in the two comparative analyses. (**B**) UpSet plot displaying the number of DE miRNAs that are co-regulated (up or down) and specifically regulated (up or down) in the Pi-vs.-PF and CK-vs.-Foc comparisons. (**C**) Heatmap depicting 24 DE miRNAs co-downregulated in Pi-vs.-PF and CK-vs.-Foc, as well as those specifically downregulated in Pi-vs.-PF. (**D**) qRT-PCR results of the selected miRNAs and their corresponding target genes. Conducted significance analysis using the ANOVA method (*p* < 0.05). (**E**) GO analysis of the target genes of 17 Pi-vs.-PF specifically downregulated DE miRNAs, with a focus on the biological process (BP) enrichment results. (**F**) Co-expression network of the 17 target genes and 6 DE miRNAs, illustrating their regulatory relationships.

**Table 1 ijms-25-12446-t001:** Statistics of RNA-Seq libraries.

Sample	Total Reads	Mapped Reads	Properly Paired	Singletons	q30
MaCK_rep1	66,181,559	48,710,169 (73.60%)	39,544,642 (60.54%)	7,828,698 (11.99%)	89.03
MaCK_rep2	65,630,392	48,146,544 (73.36%)	39,001,802 (60.18%)	7,873,618 (12.15%)	89.15
MaCK_rep3	66,256,906	48,748,192 (73.57%)	39,563,180 (60.47%)	7,889,120 (12.06%)	89.27
MaFoc_rep1	69,960,898	61,843,457 (88.40%)	55,736,072 (81.37%)	3,948,941 (5.76%)	91.46
MaFoc_rep2	65,201,889	58,165,987 (89.21%)	52,432,810 (82.17%)	3,694,600 (5.79%)	91.43
MaFoc_rep3	69,061,534	61,785,802 (89.46%)	55,755,386 (82.40%)	3,970,792 (5.87%)	91.18
MaPi_rep1	64,992,270	47,105,992 (72.48%)	38,149,018 (59.48%)	7,594,512 (11.84%)	89.28
MaPi_rep2	68,569,626	54,258,430 (79.13%)	45,978,828 (68.22%)	6,563,588 (9.74%)	89.27
MaPi_rep3	72,055,583	58,179,193 (80.74%)	50,481,932 (71.22%)	5,943,396 (8.38%)	91.48
MaPF_rep1	71,828,787	51,828,268 (72.16%)	43,504,054 (61.40%)	6,763,765 (9.55%)	91.52
MaPF_rep2	69,997,255	53,172,078 (75.96%)	45,028,898 (65.25%)	6,643,943 (9.63%)	91.57
MaPF_rep3	72,136,136	53,904,514 (74.73%)	45,274,378 (63.70%)	7,026,276 (9.89%)	91.48

**Table 2 ijms-25-12446-t002:** Statistics of miRNA libraries.

Sample Name	Sequence Type	Raw Tag Count	Low Quality Tag Count	Invalid Adapter Tag Count	PolyA Tag Count	Short Valid Length	Clean Tag Count	Q20 of Clean Tag (%)	Percentage of Clean Tag (%)
CK-1	SE50	29,431,693	524,482	1,732,986	12	668,700	26,505,513	99.2	90.06
CK-2	SE50	27,694,747	396,391	941,975	23	615,557	25,740,801	99.3	92.94
CK-3	SE50	24,875,946	434,091	1,143,765	68	176,784	23,121,238	99.3	92.95
Foc-1	SE50	29,312,433	371,578	1,031,988	49	880,676	27,028,142	99.2	92.21
Foc-2	SE50	29,602,837	488,999	615,164	95	583,316	27,915,263	99.1	94.3
Foc-3	SE50	29,629,629	700,837	687,797	51	4,079,772	24,161,172	99.2	81.54
Pi-1	SE50	28,126,333	586,326	1,242,321	49	1,878,348	24,419,289	99.2	86.82
Pi-2	SE50	29,956,563	451,936	1,101,346	18	2,001,389	26,401,874	99.5	88.13
Pi-3	SE50	28,272,622	624,527	511,416	60	502,973	26,633,646	99.2	94.2
PF-1	SE50	29,700,164	454,917	476,123	80	265,134	28,503,910	99.2	95.97
PF-2	SE50	29,738,063	392,411	1,089,884	208	658,411	27,597,149	99.2	92.8
PF-3	SE50	29,646,278	917,234	818,954	113	3,026,535	24,883,442	99.3	83.93

## Data Availability

The novel findings and original data from this study are detailed in the Appendix A. For additional information or questions, please contact the corresponding authors.

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
