# Peer review of "Integrated Transcriptome and sRNAome Analysis Reveals the Molecular Mechanisms of Piriformospora indica-Mediated Resistance to Fusarium Wilt in Banana"

_ijms, 2024, doi:10.3390/ijms252212446_

Round 1
Reviewer 1 Report
Comments and Suggestions for Authors
This study investigates the potential of the mycorrhizal fungus Piriformospora indica (Pi) to enhance banana resistance to Fusarium wilt, a major disease caused by Fusarium oxysporum f. sp. cubense (Foc), which threatens global banana production. By comparing mRNA and miRNA expression in different groups of banana plants—control, Foc-inoculated, Pi-colonized, and Pi-colonized followed by Foc-inoculation—through transcriptome and sRNAome analysis, researchers identified differentially expressed genes and miRNAs related to plant growth, glutathione metabolism, and stress response. The results suggest that P. indica enhances Foc resistance by regulating key genes like glutathione S-transferase and transcription factors, such as TCP, via miRNA modulation. This research highlights the potential of P. indica in managing banana wilt disease. This study is well designed and have a complete result. I have several minor concerns as followings:
Line 225: Are there any other interesting findings across groups with different modules, aside from those observed between the MEsaddlebrown and C3 modules?
Line 233: A citation is required here.
Line 491: DESeq2 needs to be properly cited. Additionally, I noticed that several other tools are missing citations.
Line 513: The various RNA categories should be clearly specified, along with a detailed description of the methodology used to perform this step.
Line 514: The Rfam database needs to be cited.
Figure 4: Why does the 24nt category in the CK group show a noticeably higher count compared to the other groups?
Lines 281–284: Is there a global analysis that considers all miRNAs and their targets to reveal potential relationships? It might also be beneficial to summarize all the target genes regulated by each miRNA.
Author Response
Comments 1: Line 225: Are there any other interesting findings across groups with different modules, aside from those observed between the MEsaddlebrown and C3 modules?
Response 1: Thank you for your insightful question. Upon further analysis of the data across various groups and modules, we have identified additional findings that we believe are of interest. Besides the observations made between the MEsaddlebrown and C3 modules, we have discovered that the MEdarkgreen and C2 modules also show significant correlation trends (with a Pearson correlation coefficient above 0.8 and a significance level below 0.05), as depicted in Supplementary Figure S4A.
Through GO analysis of the DEGs within the MEsaddlebrown and C3 modules, as well as the MEdarkgreen and C2 modules, we observed that their enrichment results are congruent with those of the individual C3 and C2 modules, as shown in Supplementary Figure S4C. This similarity suggests a functional alignment between the MEsaddlebrown and MEdarkgreen modules with the C3 and C2 modules, respectively, which further validates the construction of the co-expression network for our study. For detailed insights, please refer to lines 235-255 in the revised manuscript.
Comments 2: Line 233: A citation is required here.
Response 2: Thank you for pointing this out. We have already added a citation here. Please see Lines 258-261 of the revised manuscript.
Comments 3: Line 491: DESeq2 needs to be properly cited. Additionally, I noticed that several other tools are missing citations.
Response 3: Thank you for pointing this out. We have included corresponding citations for the tools we have utilized.
Comments 4: Line 513: The various RNA categories should be clearly specified, along with a detailed description of the methodology used to perform this step.
Response 4: Thank you for your suggestion. We have added information to the Materials and Methods. Please see lines 578-586 of the revised manuscript.
Comments 5: Line 514: The Rfam database needs to be cited.
Response 5: Thank you for pointing this out. We have already added a citation here.
Comments 6: Figure 4: Why does the 24nt category in the CK group show a noticeably higher count compared to the other groups?
Response 6: Thank you for your valuable feedback. We acknowledge your observation regarding the 24nt category in the CK group. Typically, small RNAs range from 18 to 30 nt in length, with miRNAs primarily found at 21 or 22 nt, siRNAs at 24 nt, and piRNAs at 28-30 nt. As siRNAs are not the central focus of our research, the prominence of the 24nt small RNAs in the CK group did not receive extensive analysis initially.
To prevent any potential confusion for the readers and to maintain the focus on our study's objectives, we have decided to exclude the small RNA length distribution plots from the manuscript. This revision will ensure that the presentation remains concise and relevant to the core findings of our investigation. The manuscript has been updated accordingly, and we believe this change enhances the clarity and readability of our submission.
Comments 7: Lines 281–284: Is there a global analysis that considers all miRNAs and their targets to reveal potential relationships? It might also be beneficial to summarize all the target genes regulated by each miRNA.
Response 7: Thank you for your valuable comments, we have analyzed target genes of all miRNAs, and found that the target genes enriched in growth and development, including positive regulation of development, heterochronic, leaf morphogenesis, regulation of timing of transition from vegetative to reproductive phase, adventitious root development, leaf development, regulation of translation, anther development, leaf shaping, petal development, as well as stress response pathways (response to bacterium, regulation of anthocyanin biosynthetic process, regulation of secondary metabolite biosynthetic process) (Supplementary Figure S6). This indicates that miRNAs are closely related to plant growth and development, as well as stress responses.
Reviewer 2 Report
Comments and Suggestions for Authors
Dear Authors,
I have an honor to review manuscript entitled:” Integrated Transcriptome and sRNAome Analysis Reveals the Molecular Mechanisms of Piriformospora indica-Mediated Resistance to Fusarium Wilt in Banana” submitted to IJMS.
Authors concentrated on potential of Piriformospora indica (Pi), a mycorrhizal fungus, acting as plant resilience and nutrient assimilation as well as to fortify bananas against disease. Authors uncovered a significant enrichment of differentially expressed genes (DEGs) and DE miRNAs in pathways associated with plant growth and development, glutathione metabolism, and stress response.
Authors presented interesting results, but I suggest to underlined more deeply the novelty aspect in obtained results as well as in discussion, because the role of Piriformospora indica was investigated in different pathosystems;
In my opinion some aspects should be clarified and/or explained and the list of the most important point was added:
-figure 1 – is difficult to read even after enlargement to 200%- Please, rethink separating these figure into 2; for example 1D can be as a separate figure 2- because in current form the important details are lost;
- the reader should have a chance to compared and value plants phenotype during Foc,Pi,PF also with CK- especially when Authors underlined the phenomenon strengthens of immunity-Please, add it;
-Authors underlined the role of GST genes in plant-fungi interactions, Did these genes engaged in interaction with other plant pathogen ?
-Please, explain, what Authors have in mind presenting statement “fortifying banana root resistance to Foc TR4 infection”?
-Figure 2 – especially panel C presenting very important data – Please, underlined it more – maybe, in discussion part;
-taking into account the number of obtained results the discussion pat of manuscript is very poor – Please, enriched this part;
- Authors concluded that structural genes participated in plant stress response and developmental proceses; Is there any findings underling the role “GST, CAT, INO80, and RBF” in plant-pathogen interactions?
Author Response
Comments 1: figure 1 – is difficult to read even after enlargement to 200%- Please, rethink separating these figures into 2; for example 1D can be as a separate figure 2- because in current form the important details are lost;
Response 1: We are grateful for your insightful suggestions. We have adjusted the text size for all figures and separated the panel 1D into a separate Fig. 2 for better readability by the readers.
Comments 2: the reader should have a chance to compared and value plants phenotype during Foc, Pi, PF also with CK- especially when Authors underlined the phenomenon strengthens of immunity-Please, add it;
Response 2: We are grateful for your insightful suggestions. We have been added the phenotypes of banana plants infected with Foc and Pi as Figure 1A and supplemented corresponding contents at the line 133-135.
Comments 3: Authors underlined the role of GST genes in plant-fungi interactions, Did these genes engaged in interaction with other plant pathogen?
Response 3: Thank you for your insightful question. Indeed, the role of GST genes in plant-fungal interactions has been well-documented across a spectrum of plant species. Moreover, it is noteworthy that infections by other pathogens, including bacteria and viruses, often induce oxidative stress and subsequently upregulate the expression of GST genes. This response has been characterized in model plants such as Arabidopsis thaliana, as well as in economically important species like tomato and tobacco. To provide a more comprehensive background, we have expanded the discussion on this topic in the introduction section, as detailed on lines 67-91. This addition enriches the context of our study and highlights the broader implications of GST genes in plant-pathogen interactions.
Comments 4: Please, explain, what Authors have in mind presenting statement “fortifying banana root resistance to Foc TR4 infection”?
Response 4: We are grateful for your insightful suggestions. The intent here is that the miRNAs specifically down-regulated in samples after P. indica colonization and subsequent FocTR4 infection were, conversely, induced during sole FocTR4 infection (except miR160a-5p). Therefore, we speculate that these miRNAs may be involved in the process by which P. indica enhances the resistance of banana roots to FocTR4. We acknowledge that the previous expression was slight misrepresentation and have we made the necessary revisions.
Comments 5: Figure 2 – especially panel C presenting very important data – Please, underlined it more – maybe, in discussion part;
Response 5: We are grateful for your insightful suggestions. Panel C presents the structural genes in the C3 module that are involved in both the regulation of growth and glutathione metabolic process pathways, including ADCP1 (Ma11_g04710, Ma02_g12430), GST (Ma10_g09500, Ma09_g26590, Ma09_g26580, Ma09_g26540, Ma05_g10110, Ma04_g36090, Ma09_g26530, Ma05_g10100), YLR126C (Ma06_g31460, Ma04_g18160), ING2-like (Ma01_g20540), NSA2 (Ma00_g03000), and CYP (Ma03_g08450). The GST and CYP genes have been reported to be related to plant stress responses. GST plays a crucial role in plant-pathogen interactions by enhancing plant antioxidant capacity. ING2-like and ADCP1 are associated with histone modifications. We have described these genes in detail in the discussion section. Additionally, most of these genes are also target genes of miRNAs that are specifically downregulated in Pi vs PF, suggesting that they may be involved in the process by which P. indica enhances banana resistance to Foc TR4.
Comments 6: taking into account the number of obtained results the discussion pat of manuscript is very poor – Please, enriched this part;
Response 6: We are grateful for your insightful suggestions. We have supplemented the discussion section with an analysis of the current research findings in conjunction with previous studies. Please see Lines 383-470 of the revised manuscript.
Comments 7: Authors concluded that structural genes participated in plant stress response and developmental proceses; Is there any findings underling the role “GST, CAT, INO80, and RBF” in plant-pathogen interactions?
Response 7: Thank you for your valuable query. In our manuscript, we have identified several structural genes that are implicated in plant stress response and developmental processes, including GST, CAT, INO80, and RBF. Specifically, the gene YLR126C (Ma06_g31460, Ma04_g18160) in bananas is annotated as a glutamine amidotransferase (GAT), and NSA2 (Ma00_g03000) encodes a homolog of the ribosome biogenesis factor (RBF) NSA2. While there is a dearth of literature on the roles of these two proteins in plants, we have taken care to correct the nomenclature of these genes within the manuscript and in Figure 6.
INO80 is known to play a role in plant flowering, immune responses, and the regulation of miRNA expression, yet its functional role and molecular mechanisms in plant defense remain largely elusive. To address your query, we have expanded upon the role of GST in plant-pathogen interactions within both the introduction and discussion sections of the revised manuscript. Please refer to lines 67-91 and 442-445 for these updates.
We appreciate the opportunity to clarify the contributions of these genes to the plant's defense repertoire and have endeavored to provide a more detailed exposition in the revised manuscript.